# Age-associated accumulation of carbamylation-derived products in tissues is independent from the myeloperoxidase pathway in mice

Christine Pietrement[1,2], Anaïs Okwieka [1], Lucile Cadoret [1], Manon Doué [1], Philippe Gillery[1,3], Stéphane Jaisson [1,3]*

1 Extracellular Matrix and Cell Dynamics unit (MEDyC) UMR 7369, University of Reims Champagne-Ardenne, CNRS, Reims, France, 2 Department of Pediatrics (Nephrology unit), University Hospital of Reims, Reims, France, 3 Department of Biochemistry-Pharmacology-Toxicology, University Hospital of Reims, Reims, France

* sjaisson@chu-reims.fr

## Abstract

Carbamylation is a nonenzymatic post-translational modification that alters protein structural and functional properties and is involved in the pathogenesis of many diseases. It results from isocyanic acid binding to protein amino groups, generating carbamylation-derived products, including homocitrulline (HCit) when the reaction targets the ε-amino group of lysine residues. Isocyanic acid is produced by two major sources *in vivo*, the spontaneous dissociation of urea and the myeloperoxidase (MPO)-catalyzed conversion of thiocyanate, but their respective contribution to carbamylation is disputed in literature. Here, we compared tissue accumulation of HCit in wild-type versus MPO-deficient mice during ageing. Our results showed that the kinetics and amplitude of carbamylation were not reduced in MPO-deficient mice. Furthermore, carbamylation was intriguingly enhanced in younger MPO-deficient mice, suggesting the presence of compensatory mechanisms. These findings suggest that the MPO pathway is not necessarily required for age-associated systemic carbamylation.

## Introduction

Molecular ageing of proteins caused by nonenzymatic post-translational modifications (NEPTMs) is well recognized for its involvement in human pathophysiological processes and complications of ageing and various long-term diseases including diabetes mellitus, chronic kidney disease (CKD) and atherosclerosis [1,2].

Carbamylation is a NEPTM characterized by the irreversible binding of isocyanic acid to amino groups of proteins, peptides and amino acids. Preferential sites of carbamylation are N-terminal extremities of proteins and ε-amino groups of lysine residues. In the latter case, homocitrulline (HCit) residue, a characteristic carbamylation-derived product (CDP), is formed [3]. Carbamylation has been

**Data availability statement:** All relevant data are within the manuscript.

**Funding:** This study was supported by grants from the Committee of American Memorial Hospital (Reims, France and Boston, MA, USA), the University of Reims Champagne-Ardenne (URCA) (France), and the Centre National de la Recherche Scientifique (CNRS, France). The funders had no role in study design, data collection and analysis, decision to publish, or preparation of the manuscript.

**Competing interests:** The authors have declared that no competing interests exist.

shown to alter many structural and functional properties of proteins [3,4]. As this cumulative process is amplified with time, long-lived extracellular matrix proteins such as collagens and elastin are privileged targets [5,6]. Many experimental and clinical studies have evidenced its responsibility in the pathophysiology of CKD and atherosclerosis [7].

Apart from minor exogenous sources of cyanate brought from the environment [8], the major endogenous sources of isocyanic acid described are (i) the spontaneous dissociation of urea and (ii) the myeloperoxidase (MPO)-catalyzed transformation of thiocyanate. The "urea pathway" is a systemic process associated with ageing [9] and is significantly increased in patients with CKD due to the high concentration of urea in blood and tissues [10,11]. The "MPO pathway" has been described to occur mainly in the context of atherosclerosis [12], as this enzyme is particularly active in inflammatory processes occurring in the arterial wall [13]. That is why this pathway has been hypothesized to be particularly important in the development of atherosclerosis [14].

However, the respective importance of the urea and MPO pathways in the carbamylation process at the systemic level has not been clearly established and is usually discussed, especially in the context of chronological ageing. In this study, we have used MPO-deficient mice in order to evaluate the intensity of carbamylation in the absence of the MPO pathway during ageing, and to determine which pathway was predominant in tissue protein carbamylation at the systemic level.

## Materials and methods

### Mice

Experiments were performed in C57BL/6J wild-type (WT) and MPO$^{-/-}$ (B6.129X1-Mpotm1Lus/J) knockout (KO) mice purchased from Charles River Laboratories (Calco, Lecco, Italy). The genotype of MPO$^{-/-}$ mice was verified by PCR. Animals were fed *ad libitum* and housed in a room with constant ambient temperature under a 12-h light–dark cycle. All animal procedures were conducted and approved by the institutional animal care committee of the University of Reims-Champagne-Ardenne (registration 56) and the veterinary services in accordance with French government policies (APAFIS#4436-2016030912045528v2).

For ageing experiments, mice were randomly assigned to five groups according to their age (1, 6, 12, 18, and 24 months; n = 10–18 depending on the group). They were euthanized by cervical dislocation under xylazine [2% (wt/vol) Rompun; 6 µg/g body weight; Bayer] and ketamine (Clorketam 1000; 120 µg/g body weight; Vetoquinol SA) anesthesia.

### Blood and tissue collections

Blood was collected in heparinized tubes after cardiac puncture, centrifuged and plasma frozen at −80°C until analysis. Back skin and aorta were removed immediately after euthanasia, and frozen at −80°C until analysis.

## Tissue extraction

Tissues (~100 mg) were homogenized with 1 mL 0.5 M acetic acid in Lysing Matrix D Tubes with the FastPrep-24 System (MP Biomedicals, California, United States of America). After homogenization, samples underwent pepsin digestion (10% wt/wt) for 24 h at 37°C.

## Urea and HCit quantification

Urea was assayed in plasma using a routine urease method (Cobas, Roche Diagnostics, Meylan, France).

HCit was quantified in plasma and tissue samples using a previously described liquid chromatography coupled to tandem mass spectrometry (LC-MS/MS) method [15]. Briefly, samples were subjected to acid hydrolysis with 6 M HCl for 18 h at 110°C and hydrolysates were twice evaporated to dryness under a nitrogen stream. Dried samples were resuspended in 100 μL of 125 mM ammonium formate containing 1 μM $d_3$-HCit and 65 μM $d_8$-Lys used as internal standards and filtered using Uptidisc PTFE Filters (4 mm, 0.45 μm; Interchim). Hydrolysates were 10-fold diluted in the same buffer and subjected to LC–MS/MS analysis (TSQ Quantis™, Thermo Scientific, Villebon-sur-Yvette, France) to quantify HCit and lysine. Mass spectra and chromatograms were acquired and processed with XCalibur software, version 1.5.1. (Thermo Scientific, Villebon-sur-Yvette, France). Results were expressed as ratios to lysine content in samples extracts.

## Statistical analysis

Data are presented as mean ± standard deviation (SD). Normal distribution of the data was assessed using the Shapiro-Wilk normality test. Statistical analysis was performed using a Two-way Analysis of Variance (ANOVA). When a significant interaction or main effect was detected, Tukey's multiple comparisons test was applied to compare WT and MPO$^{-/-}$ mice at each specific time point. All statistical analyses were performed using GraphPad Prism v8 and differences were considered statistically significant for a p value below 0.05.

## Results

Plasma measurements showed a stable level of urea and HCit concentrations over time in both WT and MPO$^{-/-}$ mice (Fig 1). Urea concentrations were not significantly different between WT and MPO$^{-/-}$ mice (around 8 mmol/L). However, plasma HCit concentrations were significantly higher in MPO$^{-/-}$ mice than in WT mice at M6 and M12.

In tissues, HCit content increased progressively with age in both mice strains, both in skin and aorta (Fig 2), reaching 1.0 mmol/mol Lys and 0.5 mmol/mol Lys, respectively, in 24-month-old mice. Surprisingly, tissue HCit levels were also higher in MPO$^{-/-}$ than in WT mice, in both skin and aorta samples.

The comparison of ratios of aorta HCit to skin HCit showed a transient decrease at 6 months of age in both WT and MPO$^{-/-}$ mice (Table 1), which may be explained by the greater accumulation of HCit in the skin at that period. These ratios were thereafter stable over time in both WT and MPO$^{-/-}$ mice, indicating that HCit accumulation in these two tissues follows the same kinetics and occurs with the same amplitude independently from the animal strain. These results showed that MPO deficiency does not significantly reduce HCit accumulation in tissues, especially in aorta.

## Discussion

NEPTMs are involved in the progression of pathological conditions and diseases in the human organism. This has been well demonstrated for nonenzymatic glycation in the development of diabetes mellitus long-term complications [16]. Nonenzymatic glycation is characterized by the binding of glucose to proteins and the subsequent formation of bioactive compounds called advanced glycation end-products (AGEs). Carbamylation is another NEPTM that has more recently been identified as responsible for pathological processes. Many experimental protocols as well as clinical studies have evidenced its responsibility in the pathophysiology of CKD and atherosclerosis, but also in ageing [7].

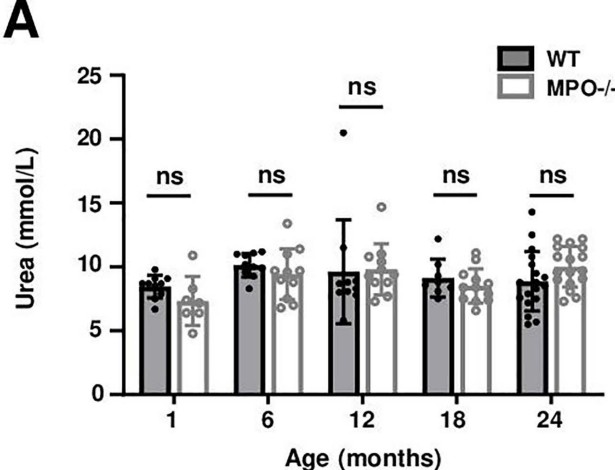

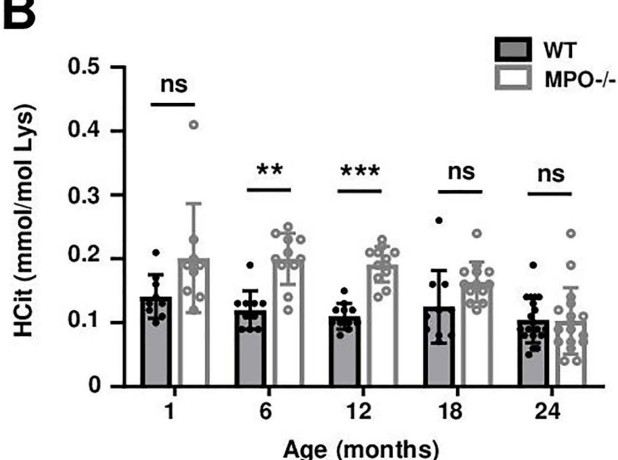

**Fig 1. Evaluation of plasma concentrations of urea (A) and HCit (B) in WT (grey bars) and MPO⁻/⁻ (open bars) mice.** Values were expressed as means ± standard deviations. Statistical significance was determined by Two-way ANOVA followed by Tukey's multiple comparisons test. **: $p < 0.01$, ***: $p < 0.0001$, (ns) non-significant.

Although based on a simple chemical reaction with amino groups, carbamylation may be initiated by different pathways depending on the mechanism of formation of the reactive component isocyanic acid. The relative importance of the two major endogenous pathways, *i.e.,* urea pathway and MPO pathway, in the carbamylation process, has not been clearly elucidated.

Previous publications in literature have emphasized the potential prevailing role of the MPO pathway, especially in the context of cardiovascular diseases, based on associations observed between inflammation and protein carbamylation. As a major evidence, correlations between carbamylated proteins (or HCit) and MPO concentrations in circulating blood have been established [12]. Other studies have not confirmed this relation [7,17,18], but this finding in the systemic milieu does not imply that the MPO pathway is not a major event at the local level in arterial walls, playing a pivotal role in the genesis and progression of atherosclerosis lesions. For example, previous studies have shown that the HCit content of high-density lipoproteins derived from atherosclerotic lesions was correlated with levels of 3-chlorotyrosine, a specific MPO oxidation product. This suggests that MPO-mediated carbamylation occurs in the vessel wall [19].

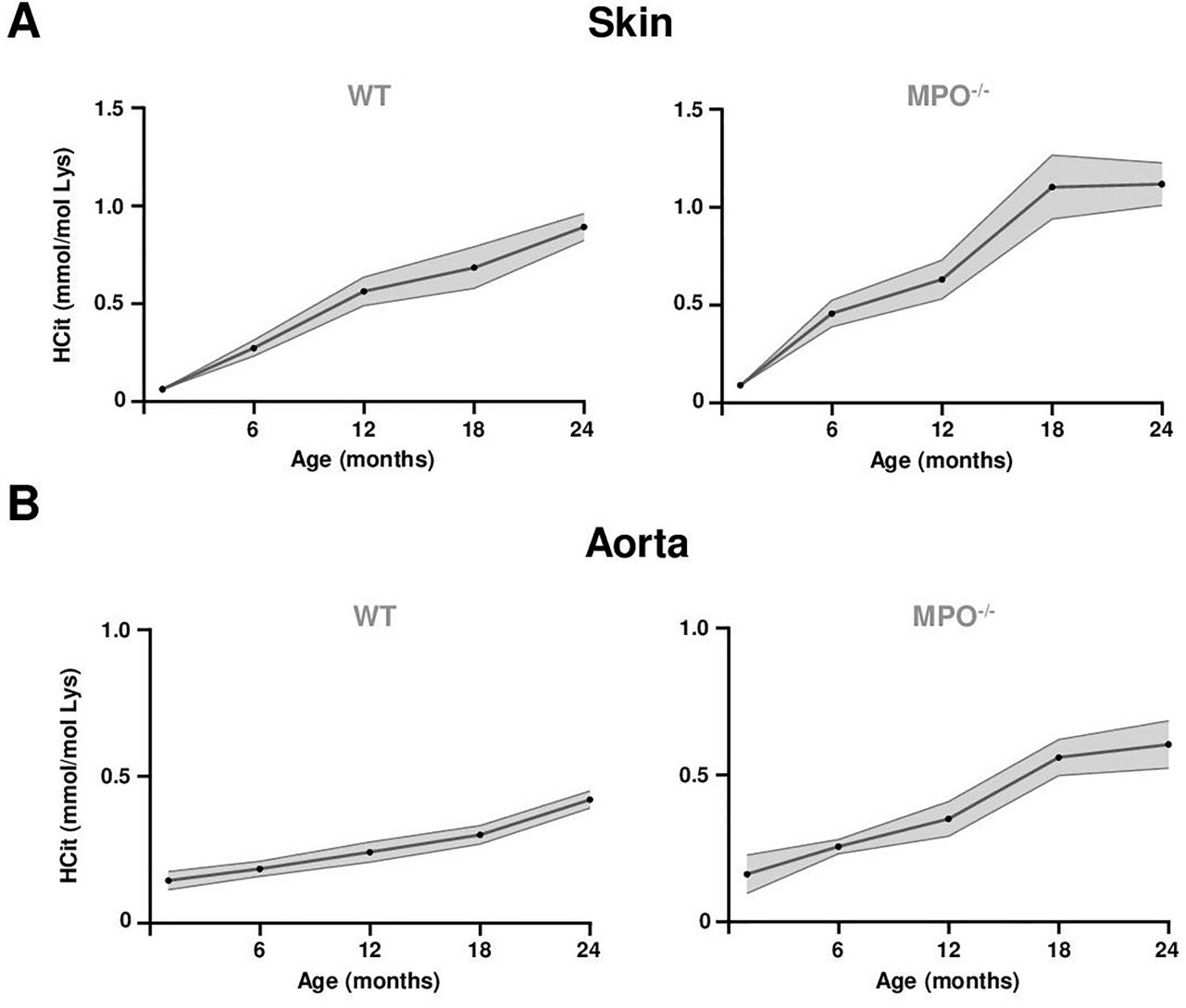

**Fig 2. HCit accumulation in skin (A) and aorta (B) of WT and MPO^-/- mice.** Dark lines represent the evolution of average HCit values at each age while grey areas represent 95% confidence intervals of the values.

**Table 1. Evolution of aorta/skin HCit ratios during ageing in wild-type and MPO^-/- mice.**

| Age (months) | | 1 | 6 | 12 | 18 | 24 |
|---|---|---|---|---|---|---|
| Aorta/skin HCit ratios | WT | 2.31 | 0.68 | 0.43 | 0.44 | 0.47 |
| | MPO^-/- | 1.77 | 0.56 | 0.56 | 0.51 | 0.54 |

The situation may be different in the context of ageing and at the systemic level. We have demonstrated previously that carbamylation is a hallmark of ageing in three species with different life expectancies, a correlation being found between age and HCit accumulation in skin and aorta [9]. The present study was designed in order to elucidate the participation of the MPO pathway in this process, which was made possible with the use of MPO deficient mice. This model allowed us to

specifically correlate urea-generated carbamylation with HCit accumulation during ageing, without participation or interference of the MPO pathway. Our results clearly showed that the kinetics of carbamylation was not reduced in MPO-deficient in comparison with WT mice. These findings suggest that the contribution of MPO in tissue protein carbamylation during chronological ageing is likely to be minimal in comparison with urea dissociation. It may be assumed that MPO contribution is an additional factor of protein carbamylation involved in the development of atherogenesis within atherosclerotic plaques [12], but its importance at a systemic level may be debatable.

A puzzling finding of the present study is the higher intensity of carbamylation rate in MPO-deficient than in WT mice. Indeed, plasma and tissue HCit values were overall significantly higher in MPO-deficient animals, whereas blood urea concentrations were similar. No obvious explanation may be given, except for the existence of compensation mechanisms. It may be hypothesized that other factors than urea concentration itself are involved in the rate of carbamylation. As it has been demonstrated that these mice exhibited a better control of lipid and carbohydrate metabolisms than WT animals [20], they may also exhibit other metabolic peculiarities. For example, a better dissociation of urea or an easier binding of isocyanic acid to proteins in an MPO-deficient context could be responsible for this apparent discrepancy. Besides, an enzyme involved in inflammatory processes (Laccase domain-containing 1, LACC1) has recently been shown to produce isocyanic acid locally by catalyzing the conversion of citrulline into ornithine [21]. This could represent a potential compensatory source of isocyanic acid independent of MPO and urea dissociation, but this hypothesis needs to be verified by further experiments. Similar intriguing situations have been described in the case of nonenzymatic glycation in humans, with the notion of genetic factors [22] generating "high and low glycation phenotypes" [23], in order to explain discrepancies in $HbA_{1c}$ formation rate at comparable blood glucose concentrations. Similar factors may be involved in carbamylation.

It is important to emphasize that our study model has some limitations, and extrapolating the results to human physiology may be difficult. Firstly, the level of MPO expression in mice can be up to ten times lower than in humans [24,25]. Similarly, the proportion of neutrophils (MPO secretors) among white blood cells is higher in humans than in mice [24,26]. When these two factors are considered together, MPO expression is lower in mice, meaning that the impact of its gene invalidation will be less significant than expected in comparison with humans.

Furthermore, the experiments were conducted under non-inflammatory conditions, whereas several human studies have shown that ageing is often associated with low-grade inflammation and increased MPO expression with age [27,28]. Finally, we quantified total HCit, including its free form as well as its protein-bound form. Therefore, we cannot rule out the possibility that certain pathways participating in the production of free HCit (particularly those involving the Carbamoyl-Phosphate Synthetases enzymes) were stimulated in $MPO^{-/-}$ mice. This would constitute another compensatory mechanism. All these limitations lead us to be cautious about the conclusions of this study and to emphasize that it is difficult to extrapolate these results to humans.

In conclusion, the MPO pathway seems to be not necessarily required for age-associated systemic carbamylation contrary to urea-induced carbamylation.

## Author contributions

**Conceptualization:** Philippe Gillery, Stephane Jaisson.

**Data curation:** Anaïs Okwieka, Lucile Cadoret, Manon Doue.

**Formal analysis:** Christine Pietrement, Lucile Cadoret, Manon Doue.

**Funding acquisition:** Christine Pietrement, Stephane Jaisson.

**Investigation:** Lucile Cadoret, Stephane Jaisson.

**Methodology:** Anaïs Okwieka, Lucile Cadoret, Manon Doue.

**Supervision:** Christine Pietrement, Philippe Gillery, Stephane Jaisson.

**Validation:** Stephane Jaisson.

**Writing – original draft:** Philippe Gillery, Stephane Jaisson.

**Writing – review & editing:** Christine Pietrement, Philippe Gillery, Stephane Jaisson.

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
