## [Decision Letter · Decision Letter 0]

9 Dec 2025

Dear Dr. JAISSON,

Thank you for submitting your manuscript to PLOS ONE. After careful consideration, we feel that it has merit but does not fully meet PLOS ONE’s publication criteria as it currently stands. Therefore, we invite you to submit a revised version of the manuscript that addresses the points raised during the review process.

The reviewers are requesting additional experiments or new experiments using Tg mice. Please consider either revising this paper or withdrawing it at this time.

We look forward to receiving your revised manuscript.

Kind regards,

Masao Tanaka

Academic Editor

PLOS One

Journal Requirements:

This study was supported by grants from the Committee of American Memorial Hospital (Reims, France and Boston, MA, USA), the University of Reims Champagne-Ardenne (URCA) (France), and the Centre National de la Recherche Scientifique (CNRS, France).

Additional Editor Comments:

The reviewers are requesting additional experiments or new experiments using Tg mice. Please consider either revising this paper or withdrawing it at this time.

Reviewers' comments:

Reviewer's Responses to Questions

**Comments to the Author**

1. Is the manuscript technically sound, and do the data support the conclusions?

Reviewer #1: Partly

Reviewer #2: Partly

2. Has the statistical analysis been performed appropriately and rigorously?

Reviewer #1: No

Reviewer #2: Yes

3. Have the authors made all data underlying the findings in their manuscript fully available?

Reviewer #1: Yes

Reviewer #2: Yes

4. Is the manuscript presented in an intelligible fashion and written in standard English?

Reviewer #1: Yes

Reviewer #2: Yes

Reviewer #1: It was previously reported that protein carbamylation is induced by MPO or urea. In this manuscript, the authors showed production of homocitrulline (carbamyl lysine) occurs in MPO KO mice. Interestingly, plasma and tissue (skin and aorta) homocitrulline levels were higher in MPO KO mice than WT although they showed the same blood urea concentration.

The things shown in this manuscript are just these, and no data to elucidate the mechanism was shown. And the clinical meaning of increased homocitrulline in MPO KO mice is not evaluated. In addition, the following concerns should be addressed.

Major

1. Statistical analyses: Did the authors employ correction of multiple comparison in figure 1A and B? If not, please use an appropriate correction method.

2. Amino acid homocitrulline is produced not only by protein carbamylation but also dysfunction of urea cycle; the latter occurs as a result of accumulation of carbamoyl phosphate. The reviewer may not understand correctly, but the method author employed just detect carbamylated protein or both carbamylated protein and amino acid homocitrulline? If amino acid is also detected, the authors should analyze the efficiency of urea cycle. And the activity of CPS1 and CPR2, which are the enzymes that produce carbamoyl phosphate synthetase, should be evaluated.

Reviewer #2: The topic oft he study is of interest. The authors clearly demonstrate that the urea pathway is responsible for the age-related accumulation of HCit in mouse tissues, independent of MPO activity. A weakness of the study is that MPO plays a minor role in murine innate immunity. The MPO content of murine neutrophils is low (~25% of human levels). Therefore, MPO-derived HCit is expected to be negligible outside of acute inflammation, which is consistent with the results.

Recommendations:

1.) MPO is not a relevant source under the low-inflammatory, low-thiocyanate conditions of aging mice. This reviewer recommends incorporating inflammation and thiocyanate stress by combining aging with or without thiocyanate supplementation and chronic, low-grade inflammation (e.g., LPS).

2.) To address the issue of low MPO activity in mice, the authors could validate their findings using hMPO-Tg models to evaluate MPO-dependent carbamylation under aging/inflammatory conditions.

3.) The study clearly underestimates the contribution of MPO to human aging/pathology, as human aging is fundamentally a chronic inflammatory disease. To draw more firm conclusions, HCit levels observed in mice should be compared to levels obtained human.

4.) The title "Age-Associated Accumulation of Carbamylation-Derived Products in Tissues Is Independent from the Myeloperoxidase Pathway" is misleading because it does not mention that this was only observed in mouse tissues.

**Do you want your identity to be public for this peer review?** For information about this choice, including consent withdrawal, please see our Privacy Policy

Reviewer #1: No

Reviewer #2: No

---

## [Author Response · Author response to Decision Letter 1]

15 Jan 2026

PONE-D-25-55644

Pietrement et al. - Age-associated accumulation of carbamylation-derived products in tissues is independent from the myeloperoxidase pathway.

Responses to reviewers:

We thank the reviewers for their insightful comments. We believe we can address each of the issues raised and make amendments to the article highlighting the limitations of the used model. The changes made to the manuscript have been highlighted in yellow.

Reviewer #1:

It was previously reported that protein carbamylation is induced by MPO or urea. In this manuscript, the authors showed production of homocitrulline (carbamyl lysine) occurs in MPO KO mice. Interestingly, plasma and tissue (skin and aorta) homocitrulline levels were higher in MPO KO mice than WT although they showed the same blood urea concentration.

The things shown in this manuscript are just these, and no data to elucidate the mechanism was shown. And the clinical meaning of increased homocitrulline in MPO KO mice is not evaluated.

Answer: The sole aim of this study was to establish a model for determining the predominant pathway of isocyanic acid formation for the carbamylation reaction of proteins. The only way to achieve this was to eliminate the MPO-dependent pathway using MPO-/- mice. Therefore, there is no mechanism to elucidate, since the cyanate formation mechanisms via these two pathways (urea- or MPO-dependent) and the conditions for protein carbamylation and its consequences are well documented in the literature. Therefore, the clinical consequences of HCit accumulation in MPO-/- mice were not evaluated, as this was beyond the scope of this study. Furthermore, these consequences can be extrapolated from those described in existing hypercarbamylation animal models including those using cyanate-supplemented drinking water (see publications by our research team and other authors).

In addition, the following concerns should be addressed.

Major

1. Statistical analyses: Did the authors employ correction of multiple comparison in figure 1A and B? If not, please use an appropriate correction method.

Answer: We thank Reviewer #1 for this helpful remark, which has enabled us to conduct more rigorous statistical analyses. As suggested, we have performed another statistical analysis of Figure 1, using a two-way ANOVA followed by a Tukey's multiple comparisons test to compare the WT and MPO-/- groups at each time point (M1 to M24). The p-values in the revised Figure 1A and 1B now account for multiple testing. The text in the Methods section, figure legends and results has been updated accordingly.

2. Amino acid homocitrulline is produced not only by protein carbamylation but also dysfunction of urea cycle; the latter occurs as a result of accumulation of carbamoyl phosphate. The reviewer may not understand correctly, but the method author employed just detect carbamylated protein or both carbamylated protein and amino acid homocitrulline?

Answer: Protein-bound HCit is a marker that can be used to determine the level of protein carbamylation. The HCit assay by LC-MS/MS used in this study and described in the referenced publication in the Methods section (Jaisson S et al. Curr Protoc. 2023;3(4):e762. doi: 10.1002/cpz1.762) includes an acid hydrolysis step to break down protein peptide bonds. This ensures that the assayed HCit reflects the total HCit content, including both the protein-bound and free forms. However, it is important to note that the proportion of free HCit relative to protein-bound HCit is negligible (less than 10%) in plasma, as demonstrated in a previous study (Jaisson et al. Anal Bioanal. Chem., 2012;402(4):1635-41, doi:10.1007/s00216-011-5619-6).

If amino acid is also detected, the authors should analyze the efficiency of urea cycle. And the activity of CPS1 and CPR2, which are the enzymes that produce carbamoyl phosphate synthetase, should be evaluated.

Answer: We find it difficult to understand what the reviewer is referring to. A CPS1/CPS2 deficiency would prevent the formation of carbamoyl phosphate and therefore limit the further formation of HCit. However, if the reviewer is considering overstimulation of the urea cycle (with an associated increase in CPS1 activity), there would be no reason for carbamoyl phosphate to accumulate, as it could be immediately metabolized by OCT enzyme. The only consequence would be an increase in uremia, which is not observed here.

HHH syndrome, which is caused by a defect in the mitochondrial ornithine transporter (ORNT1), is the only metabolic disease that could lead to increased plasma HCit concentrations. However, how could we explain how invalidation of the gene encoding MPO could result in dysfunction of this transporter?

In the manuscript, we have formulated several hypotheses to explain the difference in basal HCit concentrations between WT and MPO-/- mice (see the second paragraph on page 8), including the potential role of the LACC-1 enzyme.

Unfortunately, we cannot test this hypothesis or measure this activity (nor that of CPS-1 or CPS-2) in tissue samples because we have no longer enough samples, especially from 1- and 6-month-old mice. Furthermore, we cannot repeat an experiment that would require us to monitor the mice for two years solely to test this hypothesis.

Reviewer #2:

The topic of the study is of interest. The authors clearly demonstrate that the urea pathway is responsible for the age-related accumulation of HCit in mouse tissues, independent of MPO activity. A weakness of the study is that MPO plays a minor role in murine innate immunity. The MPO content of murine neutrophils is low (~25% of human levels). Therefore, MPO-derived HCit is expected to be negligible outside of acute inflammation, which is consistent with the results.

Answer: We thank Reviewer #2 for this interesting comment, which highlights the limitations of the model and summarizes the conclusions that can be drawn from our study. These conclusions correspond to our initial objectives and show that the proportion of protein carbamylation linked to the production of isocyanic acid from uremia is a systemic process. In contrast, the proportion related to MPO-dependent production appears to be a local process, influenced by factors such as MPO production in an inflammatory context or thiocyanate intake. In the conclusion of the manuscript, we were careful not to be too definitive about the negligible role of the MPO pathway. Instead, we mentioned its importance in localized areas, emphasizing that this depends on the context of inflammation (Wang et al., Nat Med, 2007, doi: 10.1038/nm1637).

A paragraph highlighting the limitations of this mouse model and mentioning the various criticisms made by Reviewer #2 has been added at the end of the discussion.

Recommendations:

1.) MPO is not a relevant source under the low-inflammatory, low-thiocyanate conditions of aging mice. This reviewer recommends incorporating inflammation and thiocyanate stress by combining aging with or without thiocyanate supplementation and chronic, low-grade inflammation (e.g., LPS).

Answer: We are aware of the limitations of the model we used, and the study's conclusions take these into account (we have even added a paragraph to the discussion section of the revised version of the article to emphasize this). Our study thus shows that under non-inflammatory conditions, the proportion of carbamylation due to cyanate production via MPO remains negligible compared to the urea pathway in mice.

An increase in thiocyanate intake and stimulation of low-grade inflammation by LPS would not necessarily be more representative of the physiological state and would lead to overstimulation of the MPO pathway. Such a model could therefore not replace the model used in the present study and would, at best, be complementary.

The difficulty of setting up such a model must also be taken into consideration, given the uncertainty surrounding the dietary source of thiocyanate, its mode of administration and the doses to be considered. The same would apply to the administration of LPS, which would need to be controlled to avoid excessive inflammatory stress. Finally, the time required to set up such a study (at least three years from design, monitoring of animals over two years to analysis of samples) is not compatible with the timeframe of this review process and would be disproportionate to test this hypothesis.

2.) To address the issue of low MPO activity in mice, the authors could validate their findings using hMPO-Tg models to evaluate MPO-dependent carbamylation under aging/inflammatory conditions.

Answer: We lack the expertise to generate this type of transgenic model, and there are various reasons why it would be difficult to implement it, such as controlling the level of MPO expression, establishing a controlled, constant and low-grade chronic inflammation and ensuring conditions for exogenous thiocyanate intake. Such a model would have as many limitations as, if not more than, the current model.

Furthermore, realizing this new experiment would take several years, considering the transgenic model creation, organizing mouse breeding, longitudinal follow-up over two years (aging) and analyzing the samples. This is not feasible within the timeframe allowed for reviewing an article.

3.) The study clearly underestimates the contribution of MPO to human aging/pathology, as human aging is fundamentally a chronic inflammatory disease. To draw more firm conclusions, HCit levels observed in mice should be compared to levels obtained human.

Answer: Reviewer #2 is right to highlight the probable underestimation of MPO's contribution to human ageing in our study. Several studies have highlighted increased MPO expression in this context, as well as its significant contribution to age-related and chronic diseases (Tang et al, 2009, doi: 10.1016/j.amjcard.2009.01.026; Khan eta al., 2018, doi: 10.3390/medsci6020033; Giovanni et al., 2010, doi: 10.1093/gerona/glp183; Son, et al., 2005, DOI: 10.1080/10715760500053461). We have addressed this point in the discussion paragraph that we added, which mentions the limits of the model.

Furthermore, as suggested by the reviewer, we compared HCit values in mice and humans. We had previously conducted a study evaluating HCit tissue accumulation in skin of different species, including mice and humans (Gorisse et al., PNAS, 2016, www.pnas.org/cgi/doi/10.1073/pnas.1517096113).

Below is an excerpt from this article:

As can be seen, the absolute HCit values are quite similar in mice and humans over a period of two years (values below 3 mmol HCit/mol Lys). In humans, these values increase over time, but this is due to the much longer exposure time (90 years versus 2 years). Moreover, the kinetics of accumulation are much faster in mice than in humans (see below). This means that, if mice had an equivalent life expectancy, their HCit concentrations would be higher than those of humans.

In addition, we conducted various studies in which we measured plasma concentrations of HCit either in mice or in humans (Pietrement et al, 2013, doi:10.1371/journal.pone.0082506; Mahmoudi et al., 2019, doi: 10.1515/cclm-2018-1322.; Laville et al., 2025, doi: 10.34067/KID.0000000797), reporting similar concentrations in both species.

4.) The title "Age-Associated Accumulation of Carbamylation-Derived Products in Tissues Is Independent from the Myeloperoxidase Pathway" is misleading because it does not mention that this was only observed in mouse tissues.

Answer: Reviewer #2 is right, we should have specified in the title that this study was conducted in a mouse model. The title has been amended accordingly.

---

## [Decision Letter · Decision Letter 1]

29 Jan 2026

Dear Dr. JAISSON,

The Editor judges that the experimental design was appropriate within the scope of the authors' stated objectives, and the discussion of the results is generally adequate. However, revisions are required as follows.

We look forward to receiving your revised manuscript.

Kind regards,

Masao Tanaka

Academic Editor

PLOS One

Journal Requirements:

Additional Editor Comments:

The Editor judges that the experimental design was appropriate within the scope of the authors' stated objectives, and the discussion of the results is generally adequate. Reviewer 1, perhaps placing somewhat excessive expectations on this study, also appears to have made excessive demands.

The final sentence of the abstract, ‘the same kinetics and amplitude of carbamylation in both strains,’ does not reflect the actual results. It should instead state that, intriguingly, carbamylation was enhanced in younger MPO-deficient mice, suggesting some compensatory mechanism was at work. This unexpected finding is the paper's most interesting aspect. Furthermore, the term “predominant” is also an expression that does not align with Reviewer 1's intent; it would be better to change it to “indicating that the MPO pathway is not necessarily required for age-associated systemic carbamylating.”

Finally, please add the references requested by Reviewer 2.

Reviewers' comments:

Reviewer's Responses to Questions

**Comments to the Author**

Reviewer #1: (No Response)

Reviewer #2: All comments have been addressed

2. Is the manuscript technically sound, and do the data support the conclusions?

Reviewer #1: No

Reviewer #2: Yes

3. Has the statistical analysis been performed appropriately and rigorously?

Reviewer #1: Yes

Reviewer #2: Yes

4. Have the authors made all data underlying the findings in their manuscript fully available?

Reviewer #1: Yes

Reviewer #2: Yes

5. Is the manuscript presented in an intelligible fashion and written in standard English?

Reviewer #1: Yes

Reviewer #2: Yes

Reviewer #1: The biggest problem in this manuscript is that the authors do not provide sufficient results to support the conclusion, and some comments in the discission are contradictory to each other.

1. It cannot be concluded that urea is the only important factor when MPO is removed. While authors mentioned the possibility of involvement of other carbamylating factors and introduced LACC1 as an example, they conclude that urea-induced carbamylation is important in mice during aging. This is too strong based on the same urea levels in WT and MPO KO mice in this manuscript. This conclusion can be reached after evaluating the involvement of various other “minor” carbamylation-related factors. For example, Joshi et al. showed that CPS1 is involved in histone carbamylation in collaboration with AhR (PMID: 26424795). And simply, other types of peroxidases such as Eosinophil peroxidase (EPO) may be involved in carbamylation (see PMID: 27587397).

2. The authors also concluded that MPO-induced carbamylation seems to occur in more specific locations and situations, especially in the case of local inflammation, but nothing supports this in this manuscript. The authors showed that tissue (skin and aorta) Hcit levels are increased in MPO KO mice, but this does not suggest the localization of carbamylation and involvement of inflammation. The author may say that this sentence is based on previous reports, but if so, this should not be included in the conclusion of this manuscript.

3. The authors replied that Hcit amino acid, not carbamylated protein, can be ignored, but this may not be the case in this manuscript. How can authors exclude the possibility of unexpected reaction in MPO KO mouse. Actually, unexpected results are obtained in MPO KO nice (elevated Hcit level), therefore, all kind of possibility should be evaluated. Please show the actual data or at least mention this point as a limitation.

Reviewer #2: All previous comments have been addressed. The authors conclude that MPO induced carbamylation contributes to inflammatory processes; notably, this was demonstrated in a prior study showing that chlorotyrosine - a fingerprint of MPO modification - correlates with carbamoyllysine content in HDL particles isolated from atherosclerotic lesions ([doi.org/10.1089/ars.2010.3]). The authors may wish to include this reference to strengthen their discussion.

**Do you want your identity to be public for this peer review?** For information about this choice, including consent withdrawal, please see our Privacy Policy

Reviewer #1: No

Reviewer #2: No

---

## [Author Response · Author response to Decision Letter 2]

17 Feb 2026

Editor Comments:

The Editor judges that the experimental design was appropriate within the scope of the authors' stated objectives, and the discussion of the results is generally adequate. Reviewer 1, perhaps placing somewhat excessive expectations on this study, also appears to have made excessive demands.

The final sentence of the abstract, ‘the same kinetics and amplitude of carbamylation in both strains,’ does not reflect the actual results. It should instead state that, intriguingly, carbamylation was enhanced in younger MPO-deficient mice, suggesting some compensatory mechanism was at work. This unexpected finding is the paper's most interesting aspect. Furthermore, the term “predominant” is also an expression that does not align with Reviewer 1's intent; it would be better to change it to “indicating that the MPO pathway is not necessarily required for age-associated systemic carbamylating.”

Finally, please add the references requested by Reviewer 2.

Responses to editor:

We thank the Editor for these insightful comments. We have incorporated the requested changes into the new version of the article (abstract and discussion) and added the reference suggested by Reviewer 2.

Reviewer #1:

The biggest problem in this manuscript is that the authors do not provide sufficient results to support the conclusion, and some comments in the discission are contradictory to each other.

1. It cannot be concluded that urea is the only important factor when MPO is removed. While authors mentioned the possibility of involvement of other carbamylating factors and introduced LACC1 as an example, they conclude that urea-induced carbamylation is important in mice during aging. This is too strong based on the same urea levels in WT and MPO KO mice in this manuscript. This conclusion can be reached after evaluating the involvement of various other “minor” carbamylation-related factors. For example, Joshi et al. showed that CPS1 is involved in histone carbamylation in collaboration with AhR (PMID: 26424795). And simply, other types of peroxidases such as Eosinophil peroxidase (EPO) may be involved in carbamylation (see PMID: 27587397).

Answer: We have revised the text to clarify our comments on the role of the urea pathway in systemic carbamylation (pages 8 and 9).

2. The authors also concluded that MPO-induced carbamylation seems to occur in more specific locations and situations, especially in the case of local inflammation, but nothing supports this in this manuscript. The authors showed that tissue (skin and aorta) Hcit levels are increased in MPO KO mice, but this does not suggest the localization of carbamylation and involvement of inflammation. The author may say that this sentence is based on previous reports, but if so, this should not be included in the conclusion of this manuscript.

Answer: We have followed Reviewer 1's recommendations: we have cited the reference that mentions the role of the MPO pathway in atherosclerotic plaques (Refs. 12 and 19), and we have removed this statement from the conclusion (page 9).

3. The authors replied that Hcit amino acid, not carbamylated protein, can be ignored, but this may not be the case in this manuscript. How can authors exclude the possibility of unexpected reaction in MPO KO mouse. Actually, unexpected results are obtained in MPO KO nice (elevated Hcit level), therefore, all kind of possibility should be evaluated. Please show the actual data or at least mention this point as a limitation.

Answer: We have mentioned this limitation in the new version of the article (page 9).

Reviewer #2:

All previous comments have been addressed. The authors conclude that MPO induced carbamylation contributes to inflammatory processes; notably, this was demonstrated in a prior study showing that chlorotyrosine - a fingerprint of MPO modification - correlates with carbamoyllysine content in HDL particles isolated from atherosclerotic lesions ([doi.org/10.1089/ars.2010.3]). The authors may wish to include this reference to strengthen their discussion.

Answer: We emphasized this point in the discussion and added the reference mentioned by Reviewer 2 (Ref 19).

---

## [Editor Report · Decision Letter 2]

25 Feb 2026

Age-associated accumulation of carbamylation-derived products in tissues is independent from the myeloperoxidase pathway in mice

PONE-D-25-55644R2

Dear Dr. JAISSON,

We’re pleased to inform you that your manuscript has been judged scientifically suitable for publication and will be formally accepted for publication once it meets all outstanding technical requirements.

Kind regards,

Masao Tanaka

Academic Editor

PLOS One

Additional Editor Comments (optional):

The authors have appropriately addressed the issues raised by the Editor and reviewers within the scope of this paper's objectives and revised the manuscript accordingly. Therefore, it is deemed suitable for publication.
---

## [Editor Report · Acceptance letter]

PONE-D-25-55644R2

PLOS One

Dear Dr. JAISSON,

I'm pleased to inform you that your manuscript has been deemed suitable for publication in PLOS One. Congratulations! Your manuscript is now being handed over to our production team.

Kind regards,

on behalf of

Dr. Masao Tanaka

Academic Editor

PLOS One